# Independent Validation of a Deep Learning nnU-Net Tool for Neuroblastoma Detection and Segmentation in MR Images

**DOI:** 10.3390/cancers15051622

**Published:** 2023-03-06

**Authors:** Diana Veiga-Canuto, Leonor Cerdà-Alberich, Ana Jiménez-Pastor, José Miguel Carot Sierra, Armando Gomis-Maya, Cinta Sangüesa-Nebot, Matías Fernández-Patón, Blanca Martínez de las Heras, Sabine Taschner-Mandl, Vanessa Düster, Ulrike Pötschger, Thorsten Simon, Emanuele Neri, Ángel Alberich-Bayarri, Adela Cañete, Barbara Hero, Ruth Ladenstein, Luis Martí-Bonmatí

**Affiliations:** 1Grupo de Investigación Biomédica en Imagen, Instituto de Investigación Sanitaria La Fe, Avenida Fernando Abril Martorell, 106 Torre A 7planta, 46026 Valencia, Spain; 2Área Clínica de Imagen Médica, Hospital Universitario y Politécnico La Fe, Avenida Fernando Abril Martorell, 106 Torre A 7planta, 46026 Valencia, Spain; 3Quantitative Imaging Biomarkers in Medicine, QUIBIM SL, 46026 Valencia, Spain; 4Departamento de Estadística e Investigación Operativa Aplicadas y Calidad, Universitat Politècnica de València, Camí de Vera s/n, 46022 Valencia, Spain; 5Unidad de Oncohematología Pediátrica, Hospital Universitario y Politécnico La Fe, Avenida Fernando Abril Martorell, 106 Torre A 7planta, 46026 Valencia, Spain; 6St. Anna Children’s Cancer Research Institute, Zimmermannplatz 10, 1090 Vienna, Austria; 7Department of Pediatric Oncology and Hematology, University Children’s Hospital of Cologne, Medical Faculty, University of Cologne, 50937 Cologne, Germany; 8Academic Radiology, Department of Translational Research, University of Pisa, Via Roma, 67, 56126 Pisa, Italy

**Keywords:** tumor segmentation, independent validation, external validation, neuroblastic tumors, deep learning, automatic segmentation

## Abstract

**Simple Summary:**

Tumor segmentation is a key step in oncologic imaging processing. We have recently developed a model to detect and segment neuroblastic tumors on MR images based on deep learning architecture nnU-Net. In this work, we performed an independent validation of the automatic segmentation tool with a large heterogeneous dataset. We reviewed the automatic segmentations and manually edited them when necessary. We were able to show that the automatic network was able to locate and segment the primary tumor on the T2 weighted images in the majority of cases, with an extremely high agreement between the automatic tool and the manually edited masks. The time needed for manual adjustment was very low.

**Abstract:**

Objectives. To externally validate and assess the accuracy of a previously trained fully automatic nnU-Net CNN algorithm to identify and segment primary neuroblastoma tumors in MR images in a large children cohort. Methods. An international multicenter, multivendor imaging repository of patients with neuroblastic tumors was used to validate the performance of a trained Machine Learning (ML) tool to identify and delineate primary neuroblastoma tumors. The dataset was heterogeneous and completely independent from the one used to train and tune the model, consisting of 300 children with neuroblastic tumors having 535 MR T2-weighted sequences (486 sequences at diagnosis and 49 after finalization of the first phase of chemotherapy). The automatic segmentation algorithm was based on a nnU-Net architecture developed within the PRIMAGE project. For comparison, the segmentation masks were manually edited by an expert radiologist, and the time for the manual editing was recorded. Different overlaps and spatial metrics were calculated to compare both masks. Results. The median Dice Similarity Coefficient (DSC) was high 0.997; 0.944–1.000 (median; Q1–Q3). In 18 MR sequences (6%), the net was not able neither to identify nor segment the tumor. No differences were found regarding the MR magnetic field, type of T2 sequence, or tumor location. No significant differences in the performance of the net were found in patients with an MR performed after chemotherapy. The time for visual inspection of the generated masks was 7.9 ± 7.5 (mean ± Standard Deviation (SD)) seconds. Those cases where manual editing was needed (136 masks) required 124 ± 120 s. Conclusions. The automatic CNN was able to locate and segment the primary tumor on the T2-weighted images in 94% of cases. There was an extremely high agreement between the automatic tool and the manually edited masks. This is the first study to validate an automatic segmentation model for neuroblastic tumor identification and segmentation with body MR images. The semi-automatic approach with minor manual editing of the deep learning segmentation increases the radiologist’s confidence in the solution with a minor workload for the radiologist.

## 1. Introduction

Image segmentation consists of the identification, delineation, and labeling of the voxels that belong to a region of interest [1]. In medical cancer images, this method is a cornerstone in the pipeline that enables the seamless extraction of quantitative radiomic features that can potentially be considered as imaging biomarkers if linked to clinical endpoints, as well as the assistance of image-guided measurement of treatment response. Segmentation is usually performed manually, which is a long-lasting observer-dependent process with inter and intra-observer variability [2]. In recent years, several studies have developed automatic tools for facilitating and standardizing this process [3,4]. Most of these solutions are based on deep-learning segmentation algorithms, usually built on convolutional neural networks (CNNs) [5]. A new CNN-based algorithm called nnU-Net was recently developed. In contrast to existing methods, it consists of an automatic deep learning-based segmentation framework that automatically configures itself, including preprocessing, network architecture, training, and postprocessing. It adjusts to any new dataset, outperforming most previous approaches [6,7].

External validation is a fundamental step before any artificial intelligence (AI) solution is applicable to clinical practice, checking for the reproducibility and generalizability of dealing with different patients, institutions, and scanners. This independent validation consists of testing the novel algorithm in a set of new patients to determine whether the tool works to an acceptable degree, as the solution might have been overfitted [8], usually performing worse in the external validation than in the internal evaluation process [9]. Despite the development of several segmentation tools in recent years, only a few were externally validated [10,11,12], while the majority are designed as proof-of-concept methodological feasibility studies [13]. A recent analytical study showed that only 6% of the publications that evaluate the performance of AI algorithms for diagnostic analysis of medical images were designed with robust external validation performance [13,14]. There is a need to demonstrate adequate generalizability of AI models through external validation in independent institutions to bridge the gap between research and clinical application, requiring software as a Medical Device regulatory specifications.

Within an international research project on pediatric cancer and imaging [15], a model to detect and segment neuroblastic tumors on MR images based on the state-of-the-art deep learning architecture nnU-Net has been developed [6]. The CNN had a median DSC of 0.965 ± 0.018 (median ± Interquartile Range (IQR)) compared to manual segmentations for a training set of 106 MR sequences (cross-validation) and a Dice Similarity Coefficient (DSC) of 0.918 ± 0.067 (median ± IQR) for an internal independent validation set of 26 MR sequences [16].

The aim of this study was to perform an independent validation of the automatic deep learning architecture nnU-Net previously developed and tested, proposing a novel semiautomatic segmentation methodology. We hypothesize that the state-of-the-art deep learning framework nnU-Net has excellent performance for automatically detecting and segmenting neuroblastic tumors on MR images in a different population and MR equipment than the ones used to train the tool.

## 2. Materials and Methods

### 2.1. Participants

The data set was collected within the scope of the PRIMAGE (PRedictive In-silico Multiscale Analytics to support cancer personalized diagnosis and prognosis, Empowered by imaging biomarkers) project [15]. A pediatric radiologist with 6 years of expertise and previous experience on segmentation tasks visually reviewed all the cases from the PRIMAGE Platform to exclude those that were used for the training and internal validation of the segmentation tool [16], those images that did not have an objectifiable tumor (such as the brain in an abdominal tumor or exams after treatment with no tumor rest), exams with important artifacts, and those not orientated in the transverse plane (Figure 1). Finally, 535 MR exams from 300 children were used for the independent external validation of the segmentation tool.

This retrospective multicenter international series compilation was made with 300 pediatric patients with neuroblastic tumor diagnosis and pathological confirmation between 2002 and 2022. All the included patients had undergone an MR exam of the anatomical region of the tumor at diagnosis (n = 290) or after initial treatment with chemotherapy (n = 36, as 26 of the included patients had studied at diagnosis and after treatment). None of them was used for the development of the published segmentation neural network [16].

Most of the patients were included in two clinical trials: Society of Paediatric Oncology European Neuroblastoma Network (SIOPEN). High-Risk Neuroblastoma Study (HR-NBL1/SIOPEN) (n = 119) with patients from 12 countries, led by St. Anna Children’s Cancer Research Institute (Vienna, Austria), and SIOPEN European Low and Intermediate Risk Neuroblastoma Protocol clinical trial (LINES/SIOPEN), led by La Fe University and Polytechnic Hospital (Valencia, Spain) (n = 107). In addition, patients who are not included in any of the above-mentioned clinical studies were also recruited from different European hospitals or clinical research institutions that collaborate in the PRIMAGE project: 62 from the German Neuroblastoma Registry NB2016 of the GPOH, led by the department of Pediatric Oncology at the University Children’s Hospital Cologne and, nine from Spain (La Fe University and Polytechnic Hospital, Hospital La Paz, Hospital Niño Jesús, Hospital Son Espases, Hospital Reina Sofía) and three from Italy (Pisa University Hospital, Istituto Gaslini). The study had the institutional Ethics Committee approvals from all involved institutions. MR images and related patients data from all partners were pseudonymized with the European Unified Patient Identity Management (EUPID) [17] system enabling a privacy-preserving record-linkage documentation connection and a secure data transition to the PRIMAGE framework.

Age at diagnosis was 18 ± 32 months (mean ± SD), range of 0 to 212 months, with a balanced gender distribution (155 girls and 145 boys). Tumor histology was neuroblastoma (263 cases), ganglioneuroblastoma (27 cases), and ganglioneuroma (10 cases).

### 2.2. MR Imaging

Automatic labeling of MR series using ML methods on DICOM metadata was applied to the dataset available on the PRIMAGE platform. From the 558 patients available at the PRIMAGE platform (October 2022), 300 with an MR exam at diagnosis were selected. Cases with an MR performed after initial treatment with chemotherapy and tumoral rests were also included.

MR images accounted for a high data acquisition variability, including different institutions with large heterogeneity in scanner vendors, protocols, and tumor location. MR exams were acquired with a 1.5 Tesla (n = 435) or 3 Tesla (n = 100) scanners, manufactured by either General Electric Healthcare (Signa Excite HDxt, Signa Explorer, Discovery, Genesis; n = 105); Siemens Medical (Aera, Skyra, Symphony, Avanto, Magneto Espree, TrioTim, Sonata, Spectra, Verio, Harmony; n = 318); Philips Healthcare (Intera, Achieva, Ingenia, Panorama; n = 109) or Canon (Titan; n = 3). All exams included a spin echo T2 weighted (T2wSE) or T2* weighted gradient echo (T2*wGE) sequence, with or without fat suppression (FS or STIR). Among the 535 used sequences, 307 were T2wSE, 11 T2*wGE FS, 176 T2wSE-FS, and 41 T2wSE-STIR. Chest images were obtained with respiratory synchronization. The Mean acquired FOV was 410 mm (median 440, range 225–500 mm).

### 2.3. Study Design

In order to achieve a reproducibility validation of the performance of the previously trained deep learning nnU-Net architecture, the heterogeneity of the dataset was increased by including all the 535 available transversal T2/T2*w MR sequences from 300 patients. The trained 3D segmentation tool was applied to all these cases within the PRIMAGE platform. The radiologist reviewed each sequence, blinded to the segmentation result, and visually localized the tumor and its boundaries. Then, she reviewed the segmentation masks in all the slices of the MR exams, visually validating the performance of the net and ensuring that the tumor was well segmented. Manual mask adjustment was performed when necessary. The time used by the radiologist for visually validating and manually editing the mask was recorded.

Both tumor location and treatment initiation were recorded. At the MR sequence level, tumor location was abdominopelvic (430 sequences: 269 in the adrenal gland, 143 abdominal, and 18 with pelvic location) or cervicothoracic (105 sequences: 95 of them thoracic, and 10 cervical). The influence of the initial chemotherapy treatment in the final segmentation results was analyzed by comparing the results at two different moments in the course of the illness, defined as time points: sequences of patients at diagnosis (n = 486) and sequences of patients after treatment (n = 49).

The dataset used for validation was completely independent of the one included in the training and tuning of the published segmentation neural network [16]. The age at diagnosis was lower in the validation set (mean of 18 ± 32 months vs. 37 ± 42 months for training-tuning). Regarding the technique, in training set, images performed after treatment were not included, while 9% of the sequences used for validation were obtained after chemotherapy treatment. Some differences were found in the manufacturer distribution, with an increase in the amount of Siemens studies included for validation and the introduction of Canon. Finally, T2wSE STIR and T2*wGE FS were included in the analysis of validation sequences. (Appendix A).

### 2.4. Convolutional Neural Network Architecture

The automatic segmentation model employed a 3D self-configuring framework for medical segmentation, nnU-Net. The net was previously trained with a cross-validation strategy developed in a cohort of 106 MR sequences of 106 patients and internally validated in a second cohort of 26 cases. The model training was performed along 1000 epochs with 250 iterations each and a batch size of 2. The loss function to optimize each iteration was based on the DSC. A z-score normalization was applied to the images [16].

This automatic segmentation algorithm was encapsulated in a docker container and integrated into the PRIMAGE platform. Then, all the cases were executed on a Kubernetes-based architecture (Quibim Precision v2.9) as batch processes. The output of these processes was the primary tumor mask, accessible from the DICOM viewer integrated into the platform for its final review.

### 2.5. Metrics

Different metrics have been described for addressing segmentation problems in order to compare the automatic segmentation with the human-defined manual ground truth. Since metrics have different properties, the following ones were selected:

The DSC, a spatial overlap index, is the most used metric in validating medical volume segmentations, allowing direct comparison between ground truth and automatic segmentations, and is also widely used to measure reproducibility [18,19,20]. It ranges from 0 (no spatial overlap between two sets) to 1 (complete perfect overlap), being expressed as:$$DSC=\frac{2TP}{2TP+FP+FN}$$

The Jaccard index (JAC) is a spatial overlap metric defined as the intersection between the masks of the two datasets divided by their union [21]:$$JAC=\frac{TP}{TP+FP+FN}$$

The Hausdorff distance (HD) is a spatial distance metric based on calculating the distances between all pairs of voxels from each mask (A or B), measuring the maximum distance of a segmentation mask (A) to the nearest point in the other mask (B). The HD measures the maximum distances between each point and is defined by [22]:
*HD(A, B) = max(h(A, B), h(B, A)),*
where *h(A,B)= max min ||a–b||*, and *a* ∈ *A*, *b* ∈ *B*.

The ROC AUC metric (sensitivity against 1-specificity) when a test segmentation is compared to ground truth is defined as [19]:$$AUC=1-\frac{1}{2}\left(\frac{FP}{FP+TN}+\frac{FN}{FN+TP}\right)$$

The true negatives have a large impact on the result since the background (normally the largest part of the image) contributes to the agreement [19]. Therefore, two additional spatial overlap-based metrics that do not include the true negatives were considered for the comprehension of the direction of the encountered errors: the modified false positive rate (FPRm) and false negative rate (FNR) of the automatic masks compared to the manually curated ground truth [16]. The FPRm considered those voxels that were identified by the automatic net as a tumor but corresponded to other structures (FP), divided by the voxels that actually corresponded to the manually curated ground truth mask (TP + FN voxels).
$$FPRm=\frac{FP}{TP+FN}$$

The FNR of the automatic segmentation to the ground truth considered voxels belonging to the tumor that the net did not include as such, divided by the ground truth voxels.
$$FNR=\frac{FN}{TP+FN}=1-Sensitivity$$

For consistency reasons, these last two metrics are expressed as 1-self, resulting in a maximum of 1 for a complete voxel-wise agreement and a minimum of 0 for a null similitude.

A simultaneous analysis (ANOVA) of the influence on the DSC of location, timepoint, magnetic field, and sequence weighting), was done.

### 2.6. Time Sparing

Time measurements (in seconds) were annotated by the radiologist when reviewing and adjusting the automatic masks (visual evaluation and manual evaluation).

## 3. Results

### 3.1. External Validation Results

From the 535 T2/T2*w MR sequences, the DSC was 0.997; 0.944–1.000 (median; Q1–Q3) and the JAC 0.996; 0.894–1.000 (median; Q1–Q3), both results being considered as excellent (Table 1). The mean ± SD HD was 7.081 ± 19.999 mm, with a small maximum distance of the automatic segmentation mask to the nearest point in the corrected mask. The mean AUC ROC was 0.930, showing an excellent agreement. The mean ± SD 1-FPRm was 0.847 ± 1.123, resulting in a high concordance between the net and the manual correction according to the non-tumor-included voxels. The mean ± SD 1-FNR was 0.917 ± 0.215, meaning that the net did not miss an important amount of tumor during the segmentation. Some examples of the segmentation performance in different cases are shown in Figure 2.

A descriptive sub-analysis attending outliers were performed. On a subpopulation analysis, 14% (74 from 535) of MR sequences had a DSC value < 0.8. These cases with high variability were visually analyzed by the radiologist to identify reasons for the low level of agreement. Complete failure was defined in 32 cases having a DSC < 0.19 (6%), reflecting an unsuccessful performance of the net: in 18 cases, the DSC was 0 as the network did not segment anything, while in 14 cases, the network segmented lymph nodes (10 cases) or structures such as non-suppressed fat (4 cases).

In 42 cases (8%), the DSC between 0.2 and 0.8 showed that the network was able to identify the tumor, but the segmentation was quite incomplete. In these cases, the mean ± SD tumor volume was 121.140 mm^3^ ± 281.210, smaller compared to the mean volume of the well-segmented tumors (194.660 mm^3^ ± 222.750). The proportion of cases after chemotherapy was larger (16 cases in this group vs. 33 cases in the well-segmented group). Six cases had cervicothoracic location, while 68 were abdominopelvic (with a slightly higher proportion of abdominopelvic cases compared to the well-segmented group). Regarding the magnetic field, 63 were acquired on a 1.5T equipment, while 11 were performed on a 3T; 40 sequences were weighted on T2 SE, four on STIR, 29 were T2 SE FS, and 1 T2 GE, showing similar proportions compared to the well-segmented group.

A descriptive analysis stratifying the results of the DSC regarding the location (abdominopelvic or cervicothoracic), timepoint (diagnosis or treatment), magnetic field strength (1.5T or 3T) and sequence weighting (T2 SE, T2 SE FS, T2 GE, STIR) was performed (Table 2). The results for the mean DSC are shown in Appendix A. A simultaneous analysis (ANOVA) of the influence on the DSC of these four factors was performed (location, timepoint, magnetic field, sequence weighting), showing that none of the factors or their interactions had a significant influence on the DSC (Appendix A). Despite not showing significant differences, the median DSC at diagnosis was 0.999, and after treatment was 0.902. The mean ± SD volume of tumors at diagnosis was larger 210.389 mm^3^ ± 227.830) than after treatment (43.467mm^3^ ± 49.745). Examples of the performance of the automatic segmentation tool and manual edition in two cases at different time points are shown in Figure 3.

### 3.2. Time Sparing

The mean time necessary for visual revision of the generated masks when no editing was necessary was 7.8 ± 7.5 s (mean ± SD). Only 136 masks required further correction, and the time required for this manual edition was 124 ± 120 s (mean ± SD). The median DSC of the 136 edited masks was 0.887 ± 0.499 (median; Q1–Q3).

## 4. Discussion

This multicentric international study, including an independent and heterogeneous data set of MR images from children with neuroblastic tumors, confirms and validates the performance of a trained ML tool to identify and delineate the lesions. The nnU-Net segmentation masks were visually validated by an experienced pediatric radiologist, with a median DSC of 0.997. Among the 535 T2/T2*w MR sequences, only 14% (74 cases with a DSC < 0.8) needed manual editing, and in only 6% (18 sequences), the tool failed to segment. The 136 manually adjusted masks had a quite good median DSC of 0.887; 0.562–0.990 (median; Q1–Q3), facilitating the radiologist tasks [16].

The automatization of the segmentation process has the added value over the manual method that it is deterministic, always providing the same outcomes given the same input images, improving repeatability [1]. The visual validation and manual editing of the automatic masks improve the usability of the method, so this semiautomatic approach is a reliable method that could be integrated into the radiology workflow even in large datasets [3,23]. Besides, this methodology implies an important reduction in the time required for segmentation tasks. In previous work, the mean time required to segment 20 neuroblastic tumor cases manually from scratch was 56 min [16]. With this approach, the mean time required for the manual edition was 124 ± 120 s (mean ± SD), resulting in a 96.5% of time reduction.

To our knowledge, this is the first study to perform an international, multicenter, and multivendor external validation of an automatic segmentation model for neuroblastic tumor identification and segmentation with body MR images. Previous studies investigated the development of semiautomatic segmentation algorithms for neuroblastic tumors on CT or MR images [24,25], making use of different imaging processing tools. Nevertheless, apart from our work, there is no preceding the literature addressing either the performance of a CNN-based solution in neuroblastic tumors nor the validation of a segmentation methodology. Despite the lack of the literature regarding neuroblastic tumor automatic segmentation architectures, the deep learning tool nnU-Net has set a new state-of-the-art in numerous segmentation problems surpassing most existing approaches [6,26,27], and as it has been proven, displays strong generalization characteristics.

Multiple AI processing tools have been developed, described, and published, but only a few have undergone external validation. Validation in an independent real-world dataset is a fundamental process in order to determine the accuracy of a model and to estimate its reproducibility and generalizability, which are essential in order to base clinical decisions on correct prediction models. Thus, this study represents an important step in the process of implementation of a new tool. Among the few preceding works that have addressed the issue of external validation of automatic segmentation algorithms in different areas, some studies have explored lesions such as glioblastoma [10], focal cortical dysplasia [11], or liver segments [28,29]. The most frequent limitation regarding external validation is the inclusion of a small number of subjects. As an example, the work addressing segmentation of craniopharyngioma on MR using a U-Net-based deep convolutional neural network performed an independent validation with 38 patients [30], while a study on the impact of manual and automatic segmentations on the radiomics models to predict the response of locally advanced rectal cancers the number of patients in the independent validation dataset was 28 cases [31]. External validation for demonstrating the robustness of newly developed radiomic tools used a series of 88 patients to predict biochemical recurrence in prostate cancer [32,33]. Our series is one of the largest using real-world cases for independent validation.

This work is a translational multicenter and multivendor study and has demonstrated the reproducibility of a segmentation CNN tool, following international recommendations for biomarker validation [34]. Other studies designed to validate deep learning segmentation algorithms are single-center, as one work aimed to quantify the skeletal muscle index and sarcopenia in metastatic renal carcinoma [35].

As expected, our study shows that the automatic tool performs worse after chemotherapy treatment (median DSC 0.902 after treatment vs. 0.999 at diagnosis), although the difference is not statistically significant. The reduction in the tumor volume and border delimitation after neuroblastoma treatment [36,37] may influence the performance of the net. Besides, previous works have demonstrated changes in radiomic features in other tumors, such as pancreatic cancer [38].

There are some limitations to this study, as only one experienced radiologist reviewed the automatic segmentations, so interobserver variability was not tested. Nevertheless, this is not a clinically relevant limitation as a high concordance between observers (with a median DSC overlap index of 0.969) performing manual segmentation in neuroblastoma has been described [16]. Another bias is that the CNN was trained with MR images in the transversal plane, and the validation has only been performed on transversal images. In case of low generalizability of the model, future work could include fine-tuning the nnU-Net tool to perform segmentations in sagittal and coronal planes.

This robust, repeatable, and automatized segmentation algorithm based on the state-of-the-art nn-Unet with final visual validation improves the consistency of the data extraction and strengthens the workflow of imaging tumor segmentation.

In conclusion, this international, multicentric, and multivendor independent validation study shows that a previously trained automatic nnU-Net tool is able to locate and segment neuroblastic tumors on T2/T2* weighted MR images in 97% of the cases, despite the body location and MR equipment characteristics. To our knowledge, this is the first study to independently validate an automatic segmentation model for neuroblastic tumor identification and segmentation with body MR images. The semi-automatic approach with minor manual editing of the DL segmentation increases the radiologist’s confidence in the solution and significantly reduces their required involvement.

## Figures and Tables

**Figure 1 cancers-15-01622-f001:**
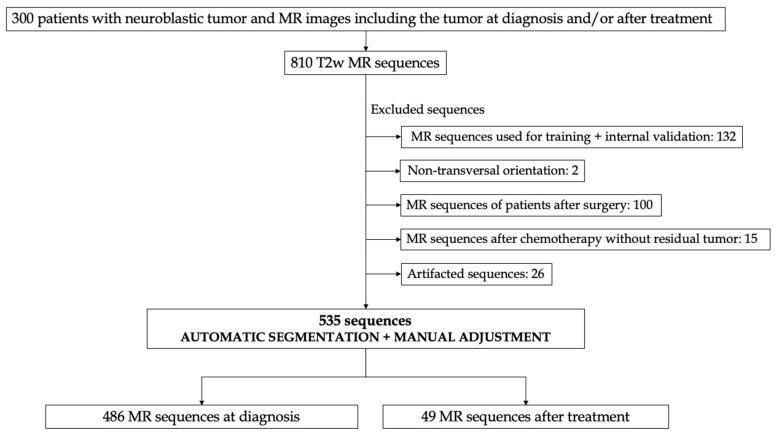
Figure depicting the study design. Transversal MR sequences were used for validation of the automatic segmentation tool for patients with neuroblastic tumors. After applying some exclusion criteria, a total amount of 300 patients, including 535 MR T2 sequences, were used for external validation, 466 sequences at diagnosis and 49 after treatment.

**Figure 2 cancers-15-01622-f002:**
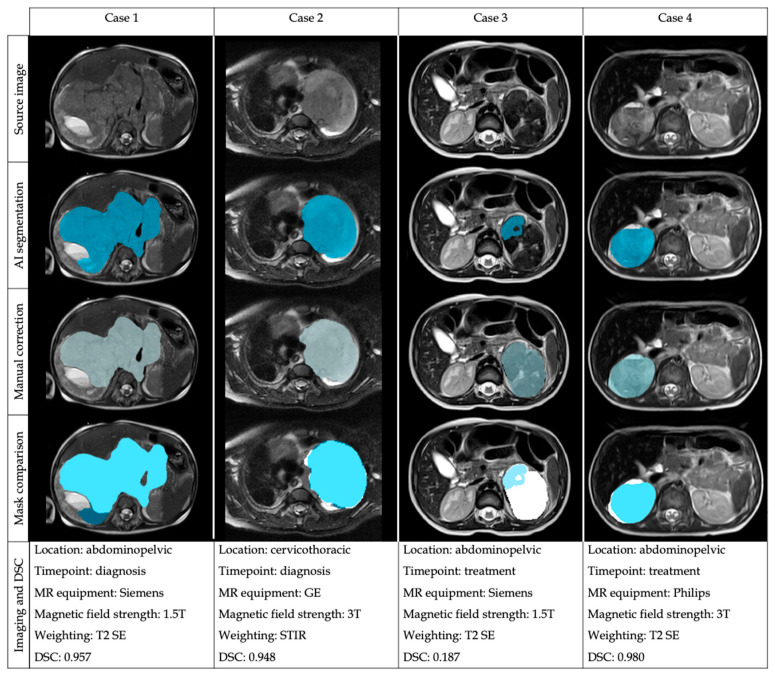
Examples of the automatic segmentation masks before and after manual edition in four different cases with heterogeneous location and imaging acquisition to show the performance of the automatic segmentation architecture and the comparison of the masks after manual correction.

**Figure 3 cancers-15-01622-f003:**
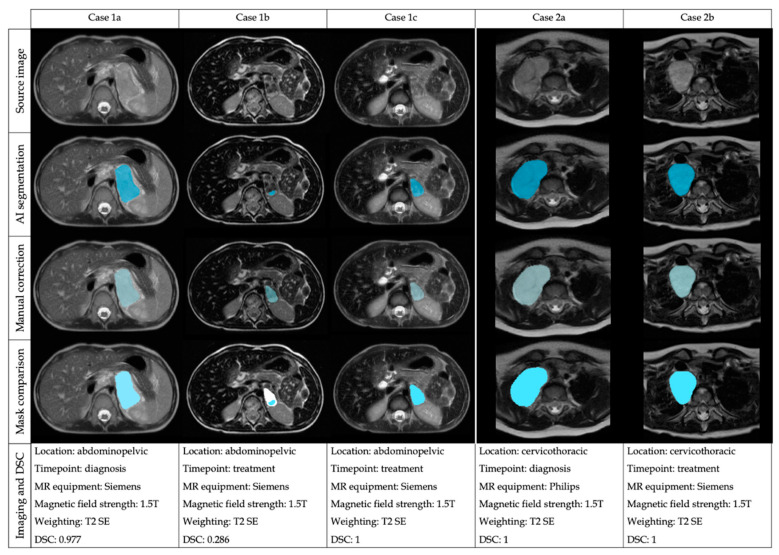
Examples of the automatic segmentation performance and manual edition in two cases (Case 1: abdominal tumor and Case 2: thoracic tumor) at different time points and performed with diverse equipment.

**Table 1 cancers-15-01622-t001:** Performance metrics for the multicentric MR studies (n = 535) considering DSC, JAC, HD, AUC ROC, 1-FPRm, and 1-FNR. From the 535 T2/T2*w MR sequences, the DSC was 0.997; 0.944–1.000 (median; Q1–Q3) and the JAC 0.996; 0.894–1.000 (median; Q1–Q3), obtaining.

	DSC	JAC	HD	AUC ROC	1-FPRm	1-FNR
Median	0.997	0.996	0.000	0.999	1.000	1.000
IQR	0.944–1.000	0.894–1.000	0.000–3.000	0.973–1.000	0.996–1.000	0.969–1.000
Mean	0.887	0.862	7.081	0.930	0.847	0.917
SD	0.262	0.279	19.999	0.191	1.123	0.215

**Table 2 cancers-15-01622-t002:** Descriptive analysis of the different metrics used: DSC, JAC, AUC ROC, 1-FPRm, and 1-FNR. Results are depicted for timepoint (diagnosis or treatment), location (abdominopelvic or cervicothoracic), magnetic field 1.5T or 3T, and sequence (T2 SE, T2 SE FS, T2 GE, STIR). DSC at diagnosis was 0.999, and after treatment was 0.902.

	DSC	JAC	HD	AUC ROC	1-FPRm	1-FNR
Tumor at diagnosis (n = 486)						
Median	0.999	0.997	0.000	0.999	1.000	1.000
Q1–Q3	0.964–1.000	0.930–1.000	0.000–2.207	0.978–1.000	0.997–1.000	0.976–1.000
Mean	0.901	0.879	7.115	0.931	0.853	0.923
SD	0.250	0.266	20.720	0.196	1.112	0.208
Tumor after chemotherapy (n = 49)						
Median	0.902	0.821	2.803	0.999	1.000	1.000
Q1–Q3	0.755–0.220	0.607–0.360	0.000–6.000	0.910–0.148	0.968–0.102	0.821–0-295
Mean	0.752	0.691	6.737	0.926	0.785	0.854
SD	0.334	0.344	10.553	0.137	1.245	0.275
Cervicothoracic (n = 105)						
Median	0.999	0.999	0.000	0.999	1.000	1.000
Q1–Q3	0.975–1.000	0.951–1.000	0.000–2.000	0.998–1.000	0.979–1.000	0.999–1.000
Mean	0.960	0.938	3.933	0.994	0.928	0.988
SD	0.109	0.145	11.712	0.023	0.212	0.046
Abdominopelvic (n = 430)						
Median	0.997	0.995	0.000	0.999	1.000	1.000
Q1–Q3	0.929–1.000	0.868–1.000	0.000–3.527	0.973–1.000	0.996–1.000	0.975–1.000
Mean	0.869	0.843	7.849	0.923	0.814	0.916
SD	0.284	0.300	21.484	0.208	1.251	0.222
1.5T (n = 434)						
Median	0.998	0.996	0.000	1.000	1.000	1.000
Q1–Q3	0.945–1.000	0.896–1.000	0.000–2.979	0.968–1.000	0.996–1.000	0.961–1.000
Mean	0.879	0.855	7.372	0.925	0.841	0.911
SD	0.275	0.291	20.622	0.197	1.174	0.223
3T (n = 101)						
Median	0.995	0.990	0.000	0.999	1.000	1.000
Q1–Q3	0.943–1.000	0.892–1.000	0.000–3.000	0.989–1.000	0.997–1.000	0.982–1.000
Mean	0.918	0.888	5.944	0.951	0.871	0.942
SD	0.196	0.220	17.251	0.163	0.884	0.177
T2 SE (n = 307)						
Median	0.997	0.994	0.000	0.999	1.000	1.000
Q1–Q3	0.951–1.000	0.906–1.000	0.000–2.855	0.971–1.000	0.995–1.000	0.951–1.000
Mean	0.904	0.877	6.078	0.938	0.873	0.907
SD	0.232	0.253	17.649	0.170	0.828	0.249
T2 SE FS (n = 176)						
Median	0.998	0.997	0.000	0.999	1.000	1.000
Q1–Q3	0.928–1.000	0.865–1.000	0.000–3.577	0.973–1.000	0.997–1.000	0.983–1.000
Mean	0.849	0.827	9.737	0.907	0.769	0.917
SD	0.317	0.327	24.984	0.239	1.618	0.226
STIR (n = 41)						
Median	0.999	0.999	0.000	0.999	1.000	1.000
Q1–Q3	0.933–1.000	0.875–1.000	0.000–2.207	0.991–1.000	0.999–1.000	0.983–1.000
Mean	0.911	0.881	3.393	0.976	0.991	0.952
SD	0.210	0.241	8.908	0.081	0.026	0.162
T2 GE FS (n = 11)						
Median	0.996	0.992	0.000	0.998	1.000	1.000
Q1–Q3	0.993–1.000	0.986–1.000	0.000–0.433	0.957–1.000	0.872–1.000	0.921–1.000
Mean	0.929	0.901	7.499	0.943	0.792	0.888
SD	0.176	0.229	18.535	0.109	0.480	0.217

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
