# Peer review of "Independent Validation of a Deep Learning nnU-Net Tool for Neuroblastoma Detection and Segmentation in MR Images"

_cancers, 2023, doi:10.3390/cancers15051622_

Round 1

Reviewer 1 Report

This is a follow-up study on the author’s previous work (published in 2022 in Cancers) that trained a machine learning model that segment pediatric neuroblastic tumor. The dataset consist of an external validation cohort of 300 patients from multiple centers/countries and imaged with different MRI scanners/settings compared to 132 in the previous training study. The effectiveness of the tool was reviewed by one radiologist and time for adjusting tumor mask was recorded. While the study provides value for validating the previous model on independed patient cohort to prove its clinical potential, the impact and quality of work is inadequate and compromised by their study design and analysis performed. Specific coments below:

Are radiologists selecting tumor mask blinded to segmentation result from the machine learing tool? If not, how did the authors ensure that the radiologist’s judgement of tumor area is not influenced by the segmentation outcome by the tool? More appropriate design would be asking the radiologist to segment the tumor independently and then compare that with the outcome of the tool.

How much time does it take radiologist to segment tumor without the ML tool vs. adjusting mask in order to support the authors’ claim that “The automatization of the segmentation process has the added value over the manual method”?

Is validation result compared to that of other ML tool to prove the hypothesis that the nnU-Net architecture is better than other state of the art CNN algorithm?

Would be helpful to clarify in Introduction what makes CNN and nnU-Net superior and state-of-the-art architecture compare to other networks for tumor segmentation?

Any example with more than one loci of tumor in one view rather than one continuous piece, which would be helpful for metastasis?

Table 2. show p – values for anova comparisons

Figure 3. Case 1a seems to be a different sequence than case 1b/c. Please double check accuracy of figure legend

Are data/algorithm deposited somewhere for validation and reproducibility by other researchers?

Author Response

REVIEWER 1

This is a follow-up study on the author’s previous work (published in 2022 in Cancers) that trained a machine learning model that segment pediatric neuroblastic tumor. The dataset consist of an external validation cohort of 300 patients from multiple centers/countries and imaged with different MRI scanners/settings compared to 132 in the previous training study. The effectiveness of the tool was reviewed by one radiologist and time for adjusting tumor mask was recorded. While the study provides value for validating the previous model on independent patient cohort to prove its clinical potential, the impact and quality of work is inadequate and compromised by their study design and analysis performed. Specific comments below:

Q1. Are radiologists selecting tumor mask blinded to segmentation result from the machine learning tool? If not, how did the authors ensure that the radiologist’s judgement of tumor area is not influenced by the segmentation outcome by the tool?

We appreciate the comments on our previous work. In our study the radiologist reviewed the images blinded to the segmentation result and visually localized the tumor and its boundaries. Then, she reviewed the segmentation mask, ensuring that the tumor was well segmented and excluding or including the structures that she considered, or editing the boundaries that had been previously revised. We have rewritten the explanation of this part (2.3. Study Design, page 4 lines 170-174).

Besides, the expertise of the radiologist was ensured, as she is a pediatric radiologist with 6 years of experience in that field, with previous experience in segmentation tasks. A brief comment has been added to the manuscript (page 3, line 106-107; page 10, 330).

Q2. More appropriate design would be asking the radiologist to segment the tumor independently and then compare that with the outcome of the tool.

This manuscript validates a previously developed deep-learning (DL) segmentation tool. In order to validate the tool, the radiologist performed the segmentation blinded and independently from the DL tool, and a comparison between both masks was performed. Thus, the described design was implemented for the training-tuning and independent validation of the net.

The aim of this manuscript was to assess the performance of the DL tool in order to stablish a semiautomatic workflow based on visual validation and manual edition of the automatically obtained segmentation masks. Given that manual segmentation is a time-consuming process that hinders the radiologist workflow, we a demonstrated that the DL tool was reliable while reducing the radiologist involvement, and that the visual validation and manual editing of the automatic masks improves the usability of the method. A brief explanation has been added to the Discussion (page 10, line 338-339).

Q3. How much time does it take radiologist to segment tumor without the ML tool vs. adjusting mask in order to support the authors’ claim that “The automatization of the segmentation process has the added value over the manual method”?

As suggested, we have added a comparison of the mean time needed to segment neuroblastic tumors from scratch with the time required with this semiautomatic approach. This methodology implies an important reduction in the time required for segmentation tasks. In our previous work, the mean time required to segment 20 neuroblastic tumor cases manually was 56 minutes. With this approach, the mean time required for manual edition was 124 s (mean±SD), resulting in a 96.5% of time reduction. (Page 10, lines 340-344).

Q4. Is validation result compared to that of other ML tool to prove the hypothesis that the nnU-Net architecture is better than other state of the art CNN algorithm?

In the case of neuroblastic tumors, there is no previous literature besides our previous publication addressing neither the performance of a CNN-based solution nor the validation of a segmentation methodology to compare the results with. Some studies have explored the development of semiautomatic segmentation algorithms. They have been performed on Computed Tomography (CT) or MR images, making use of mathematical morphology, fuzzy connectivity and other imaging processing tools, but any previous literature has demonstrated the performance of a CNN-based solution in neuroblastic tumor. We have added an explanation of this point (page 10, line 347-351).

Despite the lack of literature regarding neuroblastic tumor automatic segmentation architectures, the deep learning tool nnU-Net has set a new state-of-the-art in numerous segmentation problems surpassing most existing approaches, as demonstrated by Isensee et al. on 11 international biomedical image segmentation challenges comprising 23 different datasets and 53 segmentation tasks. (F. Isensee, P. F. Jaeger, S. A. A. Kohl, J. Petersen, y K. H. Maier-Hein, «nnU-Net: a self-configuring method for deep learning-based biomedical image segmentation», Nat. Methods, vol. 18, n.o 2, pp. 203-211, feb. 2021, doi: 10.1038/s41592-020-01008-z). We have reformulated this part on the Discussion (page 10, line 351-355).

Q5. Would be helpful to clarify in Introduction what makes CNN and nnU-Net superior and state-of-the-art architecture compare to other networks for tumor segmentation?

We have added a brief comment to make it clearer (Page 2, line 69). In contrast to existing methods, it consists of an automatic deep learning-based segmentation framework that automatically configures itself, including preprocessing, network architecture, training and postprocessing. It adjusts to any new dataset, outperforming most previous approaches.

Q6. Any example with more than one loci of tumor in one view rather than one continuous piece, which would be helpful for metastasis?

In our previous work we trained the automatic segmentation architecture excluding  metastases and lymph nodes separated from the tumor. The net has been trained only to segment the primary tumor. We have obtained very good results, as the tool did not segment liver metastases, and only in 14 cases among the 535 sequences the network segmented lymph nodes instead of the primary tumor. We chose to include different heterogeneous figures regarding location, magnetic field, T2 sequence and timepoint, as we believe these are the main factors that may impact the segmentation performance, rather than the presence of metastases.  This approach was designed to provide a reproducible, robust and universal model. Based on this statement, as far as it is possible we would like to maintain the figures as we believe they demonstrate the heterogeneity of the included sequences.

Q7. Table 2. show p – values for anova comparisons

The ANOVA comparisons of the influence on the DSC of these factors with p-values are included on Table S3 (Supplementary material). Table 2 consists of a descriptive analysis of the different metrics used with stratified results, and depicts only the description of the metrics regarding the four studied factors (location, timepoint, magnetic field, sequence weighting).

Q8. Figure 3. Case 1a seems to be a different sequence than case 1b/c. Please double check accuracy of figure legend

We have checked both studies and checked that they are both T2 SE, although one of them had a higher TE and TR than the other. 

Q9. Are data/algorithm deposited somewhere for validation and reproducibility by other researchers?

Both the data and the automatic segmentation model are available on request by contacting the PRIMAGE project, whose goal is the creation of a cloud-base clinical decision support system for the management of two pediatric cancers, neuroblastoma and DIPG.

Reviewer 2 Report

The manuscript sounds technically average; however, I have following concerns should be addressed before any decision.  

1.      Please explain in your captions of figure and title of table, why are these tables or figures necessary in your paper? What are the purposes and what are the message you want to deliver via these figures and tables?

2.      The current metrics might not be sufficient to judge the performance of the model holistically. Please enhance the result analysis part of your paper.

3.      The existing literature should be classified and systematically reviewed, instead of being independently introduced one-by-one.

4.       In the introduction section, the motivations of the proposed access control model must be included in detail. The section numbering must be changed in the paper organization paragraph.

5.      The abstract is too general and not prepared objectively. It should briefly highlight the paper's novelty as what is the main problem, how has it been resolved and where the novelty lies?

6.      The 'conclusions' are a key component of the paper. It should complement the 'abstract' and normally used by experts to value the paper's engineering content. In general, it should sum up the most important outcomes of the paper. It should simply provide critical facts and figures achieved in this paper for supporting the claims.

7.      For better readability, the authors may expand the abbreviations at every first occurrence.

8.      The author should provide only relevant information related to this paper and reserve more space for the proposed framework.

9.      The theoretical perceptive of all the models used for comparison must be included in the literature.

10.   What are the real-life use cases of the proposed model? The authors can add a theoretical discussion on the real-life usage of the proposed model.

11.   The related works section is very short and no benefits from it. I suggest increasing the number of studies and add a new discussion there to show the advantage.  

12.   The descriptions given in this proposed scheme are not sufficient that this manuscript only adopted a variety of existing methods to complete the experiment where there are no strong hypothesis and methodical theoretical arguments. Therefore, the reviewer considers that this paper needs more works.

13.   Key contribution and novelty has not been detailed in manuscript. Please include it in the introduction section

Author Response

REVIEWER 2

The manuscript sounds technically average; however, I have following concerns should be addressed before any decision.  

Q1. Please explain in your captions of figure and title of table, why are these tables or figures necessary in your paper? What are the purposes and what are the message you want to deliver via these figures and tables?

As suggested by the reviewer, we have modified the captions of Figures 1 and 2, and Tables 1, 2 and S3 to make clearer the message we want to deliver and to make it easier to understand the results.

Q2. The current metrics might not be sufficient to judge the performance of the model holistically. Please enhance the result analysis part of your paper.

Comparing segmentation masks to assess the process quality is an important step in quantitative imaging. One of the main challenges on evaluating segmentation performance is metric selection. The different described metrics, have different strengths.

In our validation manuscript we have used six main metrics: Dice Similarity Coefficient (DSC), Jaccard index (JAC), Hausdorff Distance (HD), AUC ROC, modified false positive rate (FPRm) and false negative rate (FNR), following the same process as our previous discovery paper. These metrics evaluate the performance of the model attending to different aspects providing important information on the performance of the segmentations.

The DSC, a spatial overlap index, is the most used metric in validating medical volume segmentations, allowing direct comparison between ground truth and automatic segmentations, also widely used to measure reproducibility. The JAC is also a spatial overlap metric defined as the intersection between the masks of the two datasets divided by their union, also widely used on segmentation problems. The HD is a spatial distance metric based on calculating the distances between all pairs of voxels from each mask (A or B), measuring the maximum distance of a segmentation mask (A) to the nearest point in the other mask (B). It is sensitive to outliers and to point positions. The ROC AUC metric (sensitivity against 1-specificity) is a probabilistic based metric, also widely used in segmentation problems. Two additional spatial overlap-based metrics were developed and considered for the comprehension of the direction of the encountered errors: the modified false positive rate (FPRm) and false negative rate (FNR) of the automatic masks compared to the manually curated ground truth. These metrics did not include the true negatives, as they have a large impact on the result, since the background (normally the largest part of the image) contributes to the agreement. The FPRm considered those voxels that were identified by the automatic net as tumor but corresponded to other structures (FP), divided by the voxels that actually corresponded to the manually curated ground truth mask (TP + FN voxels). The FNR of the automatic segmentation to the ground truth considered voxels belonging to the tumor that the net did not include as such, divided by the ground truth voxels.

These included different metrics are frequently used as segmentation metrics. They allow the evaluation of the validation model with a similar approach as our preliminary paper (D. Veiga-Canuto et al., «Comparative Multicentric Evaluation of Inter-Observer Variability in Manual and Automatic Segmentation of Neuroblastic Tumors in Magnetic Resonance Images», Cancers, vol. 14, n.o 15, p. 3648, jul. 2022, doi: 10.3390/cancers14153648) and provide different perspectives (overlap metrics, distance metrics, probabilistic metrics).

Q3. The existing literature should be classified and systematically reviewed, instead of being independently introduced one-by-one.

We appreciate the suggestion, we have organized the Discussion differently to make a more systematic approach.

Q4. In the introduction section, the motivations of the proposed access control model must be included in detail. The section numbering must be changed in the paper organization paragraph.

We have slightly modified the Introduction, also aligned with Q10 (Page 2, line 76).

We have followed the Cancers Instructions for Authors all over the manuscript, numbering it accordingly.

Q5. The abstract is too general and not prepared objectively. It should briefly highlight the paper's novelty as what is the main problem, how has it been resolved and where the novelty lies?

As proposed, a sentence highlighting the novelty of our work has been added to the abstract (Page 2, line 50-52).

Q6. The 'conclusions' are a key component of the paper. It should complement the 'abstract' and normally used by experts to value the paper's engineering content. In general, it should sum up the most important outcomes of the paper. It should simply provide critical facts and figures achieved in this paper for supporting the claims.

We appreciate the comment, and a sentence emphasizing the novelty of our work has been added to the conclusion (Page 11, line 425-427).

Q7. For better readability, the authors may expand the abbreviations at every first occurrence.

We have checked the abbreviations. A change has been added (Page 2 line 91).

Q8. The author should provide only relevant information related to this paper and reserve more space for the proposed framework.

We have fixed the information to the most relevant one, as suggested, by enhancing the Discussion part.

Q9. The theoretical perceptive of all the models used for comparison must be included in the literature.

The segmentation tools used in this manuscript are included in the Introduction, Materials and Methods (2.4 Convolutional Neural Network Architecture) and in the Discussion. We have added a brief explanation of the novelty and relevance of the used models (Page 10, lines 349-355). Besides, the used models are included on the References (number 6 and number 16), and all the models used for comparison are referenced:

[6] F. Isensee, P. F. Jaeger, S. A. A. Kohl, J. Petersen, y K. H. Maier-Hein, «nnU-Net: a self-configuring method for deep learning-based biomedical image segmentation», Nat. Methods, vol. 18, n.o 2, pp. 203-211, feb. 2021, doi: 10.1038/s41592-020-01008-z.

[16] D. Veiga-Canuto et al., «Comparative Multicentric Evaluation of Inter-Observer Variability in Manual and Automatic Segmentation of Neuroblastic Tumors in Magnetic Resonance Images», Cancers, vol. 14, n.o 15, p. 3648, jul. 2022, doi: 10.3390/cancers14153648

Q10.   What are the real-life use cases of the proposed model? The authors can add a theoretical discussion on the real-life usage of the proposed model.

Automatic Tumor volume and radiomic analysis might be used as clinical prediction tools, to help oncologists if the results are validated within the PRIMAGE project.

Q11.   The related works section is very short and no benefits from it. I suggest increasing the number of studies and add a new discussion there to show the advantage.  

As suggested by the reviewer, we have rewritten the Discussion part and related works. Firstly, we have structured it by exposing the results and the benefits of this approach in terms of clinical impact. Then, we have added some information about studies previously done in neuroblastic tumors to address the segmentation problem. Besides, we have added some information about the nnU-net and previous related works. Finally, we have addressed the external validation problem, regarding limitations of previously published works.

Q12.   The descriptions given in this proposed scheme are not sufficient that this manuscript only adopted a variety of existing methods to complete the experiment where there are no strong hypothesis and methodical theoretical arguments. Therefore, the reviewer considers that this paper needs more works.

The hypothesis of our work is that the state-of-the-art deep learning framework nnU-Net has an ex-cellent performance for automatically detecting and segmenting neuroblastic tumors on MR images in a different population and MR equipment than the ones used to train the tool. The aim of this study was to perform an independent validation of the automatic deep learning architecture nnU-Net previously developed and internally validated, proposing a novel semiautomatic segmentation methodology. Multiple AI processing tools have been developed, described and published but only a few have undergone an external validation. Validation in an independent real-world dataset is a fundamental process in order to determine the accuracy of a model and to estimate its reproducibility and generalizability, which are essential in order to base clinical decisions on correct prediction models. Thus, this study represents an important step in the process of implementation of a new tool.

Q13.   Key contribution and novelty has not been detailed in manuscript. Please include it in the introduction section.

We have added some comments emphasizing the novelty of our work in the abstract, introduction and conclusion (Page 2, line 50-52), (Page 3, line 96), (Page 11, line 425-427), as this is the first study to validate an automatic segmentation model for neuroblastic tumor identification and segmentation with body MR images.

Reviewer 3 Report

The ms by Veiga-Canuto et al. presents the process of independent external validation of a deep learning nnU-Net tool for neuroblastoma detection and segmentation in MR. The analysis is well designed in the context of appropriate number of analyzed images, radiologist control, implemented statistical and mathematical methods, as well as results analysis and presentations.

However, besides being properly done, the ms and research presents nothing more than external validation of "previously trained fully automatic nnU-Net CNN algorithm". Indepenent validation should be regarded as a step in implementation of a new tool, not giving it a rank of sophisticated research.

Author Response

REVIEWER 3

The ms by Veiga-Canuto et al. presents the process of independent external validation of a deep learning nnU-Net tool for neuroblastoma detection and segmentation in MR. The analysis is well designed in the context of appropriate number of analyzed images, radiologist control, implemented statistical and mathematical methods, as well as results analysis and presentations.

However, besides being properly done, the ms and research presents nothing more than external validation of "previously trained fully automatic nnU-Net CNN algorithm". Independent validation should be regarded as a step in implementation of a new tool, not giving it a rank of sophisticated research.

We really appreciate the positive feedback received and the suggestion made to improve the manuscript quality. We have added a brief statement with your suggestion: (Thus, this study represents an important step in the process of implementation of a new tool. Page 10, line 335). As we described in the manuscript, there are a lot of models that are developed but never validated. As such, independent assessment and validation of any developed tool and results remains crucial to the scientific process.

Round 2

Reviewer 1 Report

The authors addressed my comments adequately.

Reviewer 2 Report

No further comments.